# Scaling Gaussian Processes with Derivative Information Using Variational Inference

**Misha Padidar [1], Xinran Zhu[1] Leo Huang[1],**
Jacob R. Gardner[2], David Bindel[1]
[1]Cornell University, (map454, xz584, ah839, bindel)@cornell.edu
[2]University of Pennsylvania, jacobrg@seas.upenn.edu

## Abstract

Gaussian processes with derivative information are useful in many settings where derivative information is available, including numerous Bayesian optimization and regression tasks that arise in the natural sciences. Incorporating derivative observations, however, comes with a dominating $O(N^3D^3)$ computational cost when training on $N$ points in $D$ input dimensions. This is intractable for even moderately sized problems. While recent work has addressed this intractability in the low-$D$ setting, the high-$N$, high-$D$ setting is still unexplored and of great value, particularly as machine learning problems increasingly become high dimensional. In this paper, we introduce methods to achieve fully scalable Gaussian process regression with derivatives using variational inference. Analogous to the use of inducing values to sparsify the labels of a training set, we introduce the concept of inducing directional derivatives to sparsify the partial derivative information of a training set. This enables us to construct a variational posterior that incorporates derivative information but whose size depends neither on the full dataset size $N$ nor the full dimensionality $D$. We demonstrate the full scalability of our approach on a variety of tasks, ranging from a high dimensional stellarator fusion regression task to training graph convolutional neural networks on Pubmed using Bayesian optimization. Surprisingly, we find that our approach can improve regression performance even in settings where only label data is available.

## 1 Introduction

Gaussian processes (GPs) are a popular tool for probabilistic machine learning, widely used in scenarios where uncertainty quantification for regression is necessary [27, 38, 14]. When used for Bayesian optimization (BO) [18, 30], or in some regression settings found in the physical sciences like estimation of arterial wall stiffness, *derivative information* may be available [37, 34]. In these settings, we have not only noisy function values $y = f(\mathbf{x}) + \epsilon$ but also noisy gradients $\nabla \mathbf{y} = \nabla_{\mathbf{x}} f(\mathbf{x}) + \epsilon$ at some set of training points $\mathbf{X} \in \mathbb{R}^{N \times D}$. On paper, GPs are ideal models in these settings, because they allow for training on both labels $\mathbf{y}$ and gradients $\nabla \mathbf{y}$ in closed form.

Though analytically convenient, Gaussian process inference with derivative information scales poorly: computing the marginal log likelihood and predictive distribution for an exact GP in this setting requires $O(N^3D^3)$ time and $O(N^2D^2)$ memory. Recent work has addressed this scalability in certain settings, e.g. for many training points in a low-dimensional space [5] or for few training points in a high-dimensional space [3]. Despite these advances, training and making predictions for a GP with derivatives remains prohibitively expensive in regimes where both $N$ and $D$ are on the order of hundreds or even thousands.

We introduce a novel method to scale Gaussian processes with derivative information using stochastic variational approximations. We show that the expected log likelihood term of the Evidence Lower

35th Conference on Neural Information Processing Systems (NeurIPS 2021).

Bound (ELBO) decomposes as a sum over both training labels and individual partial derivatives. This lets us use stochastic gradient descent with minibatches comprised of arbitrary subsets of both label and derivative information. Just as variational GPs with inducing points replace the training label information with a set of learned *inducing values*, we show how to sparsify the derivative information with a set of *inducing directional derivatives*. The resulting algorithm requires only $O(M^3 p^3)$ time per iteration of training, where $M \ll N$ and $p \ll D$.

We demonstrate the quality of our approximate model by comparing to both exact GPs with derivative information and DSKI from [5] on a variety of synthetic functions and a surface reconstruction task considered by [5]. We then demonstrate the full scalability of our model on a variety of tasks that are well beyond existing solutions, including training a graph convolutional neural network [16] on Pubmed [28] with Bayesian optimization and regression on a large scale Stellarator fusion dataset with derivatives. We then additionally show that, surprisingly, our variational Gaussian process model augmented with inducing directional derivatives can achieve performance improvements in the regression setting even when no derivative information is available in the training set.

## 2 Background

In this section we review the background on Gaussian processes (GP) (Section 2.1), Gaussian processes with derivative observations (Section 2.2), and variational inference inducing point methods for training scalable Gaussian processes (Section 2.3).

**Derivative notation.** Throughout this paper for compactness we abuse notation slightly and use $\partial_j \mathbf{y}_i$ to refer to the $j$th element of $\nabla \mathbf{y}_i$. In this particular case, this would correspond to the partial derivative observation in dimension $j$ for training example $\mathbf{x}_i$. We also use $\partial_{\mathbf{v}} \mathbf{y}_i$ to refer to the directional derivative in the direction $\mathbf{v}$, i.e. $\nabla \mathbf{y}_i^\top \mathbf{v}$.

### 2.1 Gaussian processes

A Gaussian process (GP) is a distribution over functions $f \sim \mathcal{GP}(\mu(\mathbf{x}), k(\mathbf{x}, \mathbf{x}'))$ specified by mean and covariance function $\mu, k$ [23]. Given data points $X = \{\mathbf{x}_1, ..., \mathbf{x}_N\}$ and function observations $\mathbf{f} = \{f(\mathbf{x}_1), ..., f(\mathbf{x}_N)\}$, placing a GP prior assumes the data is normally distributed with $\mathbf{f} \sim \mathcal{N}(\mu_X, K_{XX})$ where $\mu_X$ is the vector of mean values at $X$ and $K_{XX} \in \mathbb{R}^{N \times N}$ is a covariance matrix. Conditioning on noisy observations $\mathbf{y} = \mathbf{f} + \epsilon$ where $\epsilon \sim \mathcal{N}(0, \sigma^2 I)$ induces a posterior distribution $p(\mathbf{f}^*|\mathbf{y})$ over the value of $f$ at points $\mathbf{x}^*$, which is Gaussian with mean $\mu(\mathbf{x}^*) - K_{\mathbf{x}^* X}(K_{XX} + \sigma^2 I)^{-1}(\mathbf{f} - \mu_X)$ and covariance $k(\mathbf{x}^*, \mathbf{x}^*) - K_{\mathbf{x}^* X}(K_{XX} + \sigma^2 I)^{-1} K_{X\mathbf{x}^*}$. Thus, standard GP inference takes $O(N^3)$ time. Hyperparameters such as $\sigma, \theta$ are generally estimated by Maximum Likelihood. The log marginal likelihood

$$\mathcal{L}(X, \theta, \sigma | \mathbf{y}) = -\frac{1}{2}(\mathbf{y} - \mu_X)^T (K_{XX} + \sigma^2 I)^{-1}(\mathbf{y} - \mu_X) - \frac{1}{2} \log |K_{XX} + \sigma^2 I| - \frac{n}{2} \log(2\pi) \quad (1)$$

can be optimized with methods like BFGS [19] at a complexity of $O(N^3)$ flops per iteration.

### 2.2 Gaussian processes with derivatives

GPs can leverage derivative information to enhance their predictive capabilities. Notably, as differentiation is a linear operator, the derivative of a GP is a GP [20]. Derivative observations can then be naturally included in a GP by defining a multi-output GP over the tuple of function observations and partial derivative observations $(\mathbf{y}, \nabla \mathbf{y})$ [24]. The GP has mean and covariance functions

$$\mu^\nabla(\mathbf{x}) = \begin{bmatrix} \mu(\mathbf{x}) \\ \nabla_{\mathbf{x}} \mu(\mathbf{x}) \end{bmatrix}, \qquad k^\nabla(\mathbf{x}, \mathbf{x}') = \begin{bmatrix} k(\mathbf{x}, \mathbf{x}') & \left(\nabla_{\mathbf{x}'} k(\mathbf{x}, \mathbf{x}')\right)^T \\ \nabla_{\mathbf{x}} k(\mathbf{x}, \mathbf{x}') & \nabla^2 k(\mathbf{x}, \mathbf{x}') \end{bmatrix}. \quad (2)$$

While including partial derivative observations can enhance prediction of $f$, and vice versa, a price is paid in the computational cost, as training and inference of GPs with derivatives scale as $O(N^3 D^3)$. This scalability issue has been addressed in the low $D$ setting, and is discussed in section 3.

### 2.3 Stochastic Variational Gaussian Processes

Inducing point methods [29, 22, 33, 10] achieve scalability by introducing a set of *inducing points*: an "artificial data set" of points $\mathbf{Z} = [\mathbf{z}_j]_{j=1}^M$ with associated *inducing values*, $\mathbf{u} = [u_j]_{j=1}^M$. Stochastic

Variational Gaussian Processes (SVGP) [9] augment the GP prior $p(\mathbf{f} \mid \mathbf{X}) \to p(\mathbf{f} \mid \mathbf{u}, \mathbf{X}, \mathbf{Z})p(\mathbf{u} \mid \mathbf{Z})$ and then learn a variational posterior $q(\mathbf{u}) = \mathcal{N}(\mathbf{m}, \mathbf{S})$. Inference for an observation $\mathbf{y}^*$ at $\mathbf{x}^*$ takes time $O(M^3)$:

$$q(\mathbf{y}^*) = \mathcal{N}(y^*; K_{\mathbf{x}^* Z} K_{ZZ}^{-1} \mathbf{m}, \sigma_{\mathbf{f}}(\mathbf{x}^*)^2 + \sigma^2) \tag{3}$$

where $\sigma_{\mathbf{f}}(\mathbf{x})^2 = K_{\mathbf{xx}} - K_{\mathbf{x}Z} K_{ZZ}^{-1} K_{Z\mathbf{x}} + K_{\mathbf{x}Z} K_{ZZ}^{-1} S K_{ZZ}^{-1} K_{Z\mathbf{x}}$ is the data-dependent variance. Using Jensen's inequality and the variational ELBO [10, 11], SVGP develops a loss that is separable in the training data and amenable to stochastic gradient descent (SGD) [25], as the Kullback-Leibler (KL) divergence regularization only depends on $\mathbf{u}$

$$\text{ELBO}_{\text{SVGP}} = \sum_{i=1}^{N} \left\{ \log \mathcal{N}(y_i | \mu_{\mathbf{f}}(\mathbf{x}_i), \sigma^2) - \frac{\sigma_{\mathbf{f}}(\mathbf{x}_i)^2}{2\sigma^2} \right\} - \text{KL}\left[ q(\mathbf{u}) || p(\mathbf{u}) \right]. \tag{4}$$

In equation (4), $\mu_{\mathbf{f}}(\mathbf{x}_i), \sigma_{\mathbf{f}}(\mathbf{x}_i)^2$ are the predicted mean and variance, respectively. The ELBO is maximized over the variational parameters $\mathbf{m}, \mathbf{S}$ and the GP hyperparameters $\theta$. Training with SGD on mini-batches of $B$ data points brings the time per iteration to $O(BM^2 + M^3)$.

While SVGP scales well, its predictive variances are often dominated by the likelihood noise [13]. Modeling derivatives necessarily involves heteroscedastic noise, or at least different noise for the function values and gradients, which may make SVGP with a Gaussian likelihood ill-suited to the task. The Parametric Gaussian Process Regressor (PPGPR) achieves heteroscedastic modeling by using the latent function variances without modifying the likelihood by symmetrizing the dependence of the loss on the data-dependent variance $\sigma_{\mathbf{f}}(\mathbf{x}_i)^2$ term

$$\text{ELBO}_{\text{PPGPR}} = \sum_{i=1}^{N} \log \mathcal{N}(y_i | \mu_{\mathbf{f}}(\mathbf{x}_i), \sigma^2 + \sigma_{\mathbf{f}}(\mathbf{x}_i)^2) - \text{KL}[q(\mathbf{u}) || p(\mathbf{u})]. \tag{5}$$

In Section 5, we evaluate our approach as an extension to both SVGP and PPGPR, and find that PPGPR gives significant performance gains.

## 3 Related Work

DSKI and DSKIP [5], derivative extensions of SKI [36] and SKIP [7], are among the first methods to address scaling Gaussian processes with derivative information in a low dimensional setting. DSKI and DSKIP approximate derivative kernels by differentiating interpolation kernels $\nabla k(\mathbf{x}, \mathbf{x}') \approx \sum_i \nabla w_i(\mathbf{x}) k(\mathbf{x}_i, \mathbf{x}')$ where $w_i(\mathbf{x})$ are interpolation weights used in SKI. Like SKI, DSKI suffers from the curse of dimensionality, and matrix-vector products cost $O(ND6^D + M \log M)$ time. DSKIP improves the dependence on $D$, but still costs $O(D^2(N + M \log M + r^3 N \log D))$ to form the approximate kernel matrices, where $r \ll N$ is the effective rank of the approximation. Thus while these methods exhibit high model fidelity, they are limited to low dimensional settings.

Recently [3] introduced an exact method for training GPs with derivatives in time $O(N^2 D + (N^2)^3)$, which improves on the naive $O(N^3 D^3)$ when $N < D$. This method is not applicable as $N$ grows moderately large, while our paper chiefly focuses on the high-$N$ and high-$D$ setting.

Bayesian optimization with derivatives was considered in [37]. Here, the authors consider conditioning on directional derivatives to achieve some level of scalability, but the dataset sizes considered never exceed $N$ of around 200 or $D$ of around 8. Distinct from their consideration of directional derivative information, we will be equipping *each inducing point* in a sparse GP model with its own set of distinct directional derivatives, allowing the model to learn derivatives in many directions in regions of space where there are a large number of inducing points.

## 4 Methods

Our goal is to enable training and inference on data sets with large $N$ and $D$ when derivatives are available. We will present our method in three steps. First, we describe a naive adaptation of stochastic variational Gaussian processes to the setting with derivatives. Second, we argue that this adaptation again scales poorly in $D$. Finally, we show that using additional sparsity on the derivatives gives us scalability in both $N$ and $D$.

## 4.1 Variational Gaussian processes with derivatives.

As described in Section 2.3, SVGP creates a dataset of *inducing points* $\mathbf{Z} = [\mathbf{z}_j]_{j=1}^M$ with labels (or *inducing values*) $\mathbf{u} = [u_j]_{j=1}^M$. Assume we are given a dataset $\mathbf{X} = [\mathbf{x}_i]_{i=1}^N$ with labels $\mathbf{y} = [y_i]_{i=1}^N$ and derivative observations $\nabla \mathbf{y} = [\nabla y_i]_{i=1}^N$. A natural extension of SVGP to this data is to augment the standard inducing dataset with *inducing derivatives*, $\nabla \mathbf{u} = [\nabla u_j]_{j=1}^M$, each of length $D$, so that each inducing point becomes a triple $(\mathbf{z}_j, u_j, \nabla u_j)$. This corresponds to a new augmented GP prior:

$$p(\mathbf{f}, \nabla \mathbf{f} \mid \mathbf{X}) \to p(\mathbf{f}, \nabla \mathbf{f} \mid \mathbf{u}, \nabla \mathbf{u}, \mathbf{X}, \mathbf{Z}) p(\mathbf{u}, \nabla \mathbf{u} \mid \mathbf{Z}). \quad (6)$$

Analogous to SVGP, we introduce a variational posterior:

$$q(\mathbf{u}, \nabla \mathbf{u}) = \mathcal{N}\left(\mathbf{m}^\nabla, \mathbf{S}^\nabla\right) = \mathcal{N}\left(\begin{bmatrix} \mathbf{m} \\ \nabla \mathbf{m} \end{bmatrix}, \begin{bmatrix} \mathbf{S} & \nabla \mathbf{S} \\ \nabla \mathbf{S}^\top & \nabla^2 \mathbf{S} \end{bmatrix}\right). \quad (7)$$

Here, $\mathbf{m}$ and $\nabla \mathbf{m}$ are trainable parameters learned by maximizing the ELBO. We abuse notation and call the second portion of the vector $\nabla \mathbf{m}$ because these variational mean parameters correspond to the $M \times D$ inducing derivative values. This also holds for the matrices $\nabla \mathbf{S}$ and $\nabla^2 \mathbf{S}$.

With this augmented variational posterior, the ELBO becomes:

$$\mathbb{E}_{q(\mathbf{f}, \nabla \mathbf{f})} \left[\log p(\mathbf{y}, \nabla \mathbf{y} \mid \mathbf{f}, \nabla \mathbf{f})\right] - \mathrm{KL}(q(\mathbf{u}, \nabla \mathbf{u}) || p(\mathbf{u}, \nabla \mathbf{u})). \quad (8)$$

Assuming the typical iid Gaussian noise likelihood for regression and expanding the first term further:

$$\mathbb{E}_{q(\mathbf{f}, \nabla \mathbf{f})} \left[\log p(\mathbf{y}, \nabla \mathbf{y} \mid \mathbf{f}, \nabla \mathbf{f})\right] = \\ \sum_{i=1}^N \mathbb{E}_{q(f_i)} \left[\log p(y_i \mid f_i)\right] + \sum_{i=1}^N \sum_{j=1}^D \mathbb{E}_{q(\partial_j f_i)} \left[\log p(\partial_j y_i \mid \partial_j f_i)\right]. \quad (9)$$

Here, we have used linearity of expectation and the conditional independence between $y_i$ and $\partial_j f_i$ given $f_i$ to show that the term of the ELBO that depends on training data decomposes as a sum over labels $y_i$ and partial derivatives $\partial_j y_i$. Thus, minibatches can contain an arbitrary subset of labels $(\mathbf{x}_i, y_i)$ and partial derivatives $(\mathbf{x}_i, \partial_j y_i)$, and the minibatch size $B$ remains independent of $N$ and $D$.

The moments of $q(\mathbf{f}, \nabla \mathbf{f}) = \int p(\mathbf{f}, \nabla \mathbf{f} \mid \mathbf{u}, \nabla \mathbf{u}) p(\mathbf{u}, \nabla \mathbf{u}) \, d\mathbf{u} \, d\nabla \mathbf{u}$ are similar to those in SVGP, but the kernel matrices have been augmented with derivatives (i.e., using the kernel $k^\nabla(\mathbf{x}, \mathbf{x}')$):

$$\mu_{\mathbf{f}, \nabla \mathbf{f}} = K_{XZ}^\nabla \left(K_{ZZ}^\nabla\right)^{-1} \mathbf{m}^\nabla, \quad \Sigma_{\mathbf{f}, \nabla \mathbf{f}} = K_{XX}^\nabla + K_{XZ}^\nabla K_{ZZ}^{\nabla -1} (\mathbf{S}^\nabla - K_{ZZ}^\nabla) \left(K_{ZZ}^\nabla\right)^{-1} K_{ZX}^\nabla. \quad (10)$$

Here, $\mathbf{K}_{XX}^\nabla$ is a $B \times B$ matrix that corresponds to a randomly sampled subset of label and partial derivative information. $\mathbf{K}_{XZ}^\nabla$ is $B \times M(D+1)$, and both $\mathbf{S}^\nabla$ and $\mathbf{K}_{ZZ}^\nabla$ are $M(D+1) \times M(D+1)$. Similarly, the KL divergence $\mathrm{KL}(q(\mathbf{u}, \nabla \mathbf{u}) || p(\mathbf{u}, \nabla \mathbf{u}))$ involves multivariate Gaussians with covariance matrices of size $M(D+1) \times M(D+1)$. As a result, the running time complexity of an iteration of training under this framework is $O(M^3 D^3)$ which, grows rapidly with dimension.

## 4.2 Variational Gaussian processes with directional derivatives.

The procedure above is deceptively expensive despite the asymptotic complexity of a single iteration. Because a minibatch of size $B$ contains an arbitrary subset of the $N$ labels and $ND$ partial derivatives rather than simply a subset of the $N$ labels, each epoch in the above procedure must process roughly $\frac{N+ND}{B}$ minibatches, rather than the usual $\frac{N}{B}$. Additionally, because $\mathbf{K}_{ZZ}^\nabla$ is of size $M(D+1) \times M(D+1)$, the above procedure is also analogous to SVGP using $M(D+1)$ inducing points rather than using $M$. While minibatch training adapts readily to $N(D+1)$ training examples, it is rare to use significantly more than 1000 inducing points, which can require specialized numerical tools to make scale even to $M = 10000$ [21]. In practice, $M(D+1)$ would rapidly result in matrices $\mathbf{K}_{ZZ}^\nabla$ that make training infeasibly slow.

To make the matrix $\mathbf{K}_{ZZ}^\nabla$ not directly scale with the input dimensionality, we replace the inducing derivatives from equation (7) with *inducing directional derivatives*. Rather than the triplet $(\mathbf{z}_i, u_i, \nabla \mathbf{u}_i)$ with $\nabla \mathbf{u}_i$ having dimension $D$, each inducing point is now equipped with a set of $p$ distinct directional derivatives $(\mathbf{z}_i, u_i, \partial_{\mathbf{V}_{i1}} u_i, ..., \partial_{\mathbf{V}_{ip}} u_i)$ in the directions $\mathbf{v}_{i1}, ..., \mathbf{v}_{ip}$. We include the inducing directions $\overline{\mathbf{V}} = [\mathbf{V}_1 \cdots \mathbf{V}_M] \in \mathbb{R}^{Mp \times D}$ as trainable parameters.

**GPs with Directional Derivatives.** Similar to how we built the derivative kernel matrix in Section 2.2, we may define a multi-output GP over an unknown function and its directional derivatives. For a point $\mathbf{z}_i$ and some direction $\mathbf{v}_i$ and another point and direction $\mathbf{z}_j$ and $\mathbf{v}_j$ the directional-derivative covariance function is:

$$k^{\partial_{v_i}\partial_{v_j}}(\mathbf{z}_i, \mathbf{z}_j) = \begin{bmatrix} k(\mathbf{z}_i, \mathbf{z}_j) & \nabla_{\mathbf{z}_j}k(\mathbf{z}_i, \mathbf{z}_j)^\top \mathbf{v}_j \\ \mathbf{v}_i^\top \nabla_{\mathbf{z}_i}k(\mathbf{z}_i, \mathbf{z}_j) & \mathbf{v}_i^\top \nabla_{\mathbf{z}_i\mathbf{z}_j}^2 K(\mathbf{z}_i, \mathbf{z}_j)\mathbf{v}_j \end{bmatrix}, \tag{11}$$

which is of size $2 \times 2$ rather than $(D+1) \times (D+1)$ as with $k^\nabla(\cdot, \cdot)$.

Given $Mp$ inducing directions $\overline{\mathbf{V}}$, $p$ per each of the $M$ inducing points, the relevant kernel matrices (1) between all pairs of inducing values and directional derivatives, $\overline{\mathbf{K}}_{ZZ}$, and (2) between all inducing values and training examples with full partial derivative observations, $\overline{\mathbf{K}}_{XZ}$, are:

$$\overline{\mathbf{K}}_{ZZ} = \begin{bmatrix} \mathbf{K}_{ZZ} & \nabla_Z \mathbf{K}_{ZZ}\overline{\mathbf{V}} \\ \overline{\mathbf{V}}^\top \nabla_Z \mathbf{K}_{ZZ} & \overline{\mathbf{V}}^\top \nabla_{ZZ}^2 \mathbf{K}_{ZZ}\overline{\mathbf{V}} \end{bmatrix}, \quad \overline{\mathbf{K}}_{XZ} = \begin{bmatrix} \mathbf{K}_{XZ} & \nabla_Z \mathbf{K}_{XZ}\overline{\mathbf{V}} \\ \nabla_X \mathbf{K}_{XZ} & \nabla_Z^2 \mathbf{K}_{XZ}\overline{\mathbf{V}} \end{bmatrix}, \tag{12}$$

the first of which has shape $M(p+1) \times M(p+1)$. Constructing $\overline{\mathbf{K}}_{ZZ}, \overline{\mathbf{K}}_{XZ}$ is inexpensive as we compute them directly from the directional derivative kernel (11), rather than computing the full gradient kernel $k^\nabla$ and multiplying by the directions $\overline{\mathbf{V}}$ which would incur a cost of $O(M^2D^2)$.

Variational inference with this model is nearly identical to inference with full inducing gradients. We define a variational posterior, this time over the $M(p+1)$ inducing values and directional derivatives:

$$q(\mathbf{u}, \partial_{\mathbf{V}}\mathbf{u}) = \mathcal{N}\left(\overline{\mathbf{m}}, \overline{\mathbf{S}}\right) \tag{13}$$

where $\overline{\mathbf{m}} \in \mathbb{R}^{M(p+1)}$ and $\overline{\mathbf{S}} \in \mathbb{R}^{M(p+1) \times M(p+1)}$. The model is trained by optimizing the variational ELBO

$$\mathbb{E}_{q(\mathbf{f}, \nabla\mathbf{f})}\left[\log p(\mathbf{y}, \nabla\mathbf{y} \mid \mathbf{f}, \nabla\mathbf{f})\right] - \mathrm{KL}(q(\mathbf{u}, \partial_{\mathbf{V}}\mathbf{u})||p(\mathbf{u}, \partial_{\mathbf{V}}\mathbf{u})). \tag{14}$$

over the variational parameters $\overline{\mathbf{m}}, \overline{\mathbf{S}}$, the inducing points $\mathbf{Z}$, the inducing directions $\overline{\mathbf{V}}$, and the hyperparameters. Because the structure of the ELBO remains unchanged, the training labels and partial derivatives can again be subsampled to form minibatches of size $B$, yielding $\overline{\mathbf{K}}_{XZ} \in \mathbb{R}^{B \times M(p+1)}$. Inference proceeds by computing $q(\mathbf{f}, \nabla\mathbf{f})$ from (10) by replacing the kernel matrices $K_{XZ}^\nabla$ and $K_{ZZ}^\nabla$ with our directional derivative variants $\overline{\mathbf{K}}_{XZ}$ and $\overline{\mathbf{K}}_{ZZ}$.

**Learning inducing directions** The above algorithm requires the selection of a set $\overline{\mathbf{V}} = [\mathbf{V}_1 \cdots \mathbf{V}_M]$ of inducing directions. In the setting where all inducing points have a shared set of $p$ global inducing directions, there is an optimal fixed choice for the inducing directions [2, 5]. However, for a variational GP with directional derivatives, sharing inducing directions does not improve scalability as the size of the kernel matrices $\overline{\mathbf{K}}_{XZ}, \overline{\mathbf{K}}_{ZZ}$ is unchanged from the case where each inducing point has $p$ distinct inducing directions. Thus we may improve model flexibility and performance by allowing *each inducing point to have distinct directions at no additional computation complexity* (see the supplementary materials for a comparison against shared inducing directions). In this case, a principled approach to setting the inducing directions is to include them as a set of $mpd$ trainable parameters, and learn them when maximizing the ELBO (14). Learning inducing directions allows nearby inducing points to balance the directions from which they capture information, and encourages the model to capture the locally most informative directions. We adopt this approach in our experiments in section 5.

**Derivative modeling with $p \ll D$.** A key feature of this framework is that it allows for the use of a different number $p$ of directional derivatives per inducing point than the number of partial derivative observations per training point. Particularly for kernel matrices involving training examples with full partial derivative information, using $p \ll D$ directional derivatives keeps the matrix dimension small and independent of $D$. Nevertheless, allowing each inducing point to have its own set of learnable directions enables the model to learn many derivative directions where necessary in the input space by placing multiple inducing points with different directions nearby. A notable case is when each inducing point $\mathbf{z}_i$ has the $p = D$ canonical inducing directions $\mathbf{V}_i = I$, through which we recover the full variational GP with derivatives as described in section 4.1.

**Complexity.** For a minibatch size $B$, when learning $p$ directional derivatives per inducing point, the matrices $\overline{\mathbf{K}}_{XZ}$ and $\overline{\mathbf{K}}_{ZZ}$ become $B \times M(p+1)$ and $M(p+1) \times M(p+1)$ respectively. As a result, the time complexity of variational GP inference with directional derivatives is $O(M^3 p^3)$. When using $p$ directions per inducing point, this is computationally equivalent to running SVGP with $p+1$ times as many inducing points. To counteract the additional matrix size, one may use the whitened formulation of variational inference [17] for GPs when computing equation (10) and use contour integral quadrature as in [21].

## 5 Experiments

In this section we evaluate the empirical performance of variational GPs with directional derivatives. In sections 5.1 and 5.2 we benchmark the performance of our method on small regression problems where we can compare to prior work such as exact GPs with derivatives and DSKI. In sections 5.3, 5.4, and 5.5 we compare to variational GPs without derivatives on high dimensional regression and Bayesian optimization (BO) tasks which are well beyond the scalability means of all prior work we are aware of. In section 5.6 we perform an ablation study to understand the effect of increasing the number of inducing directions. In section 5.7 we investigate the value of learning directional derivative information even when derivative observations are not available. All of our GP models use a constant prior and Gaussian kernel (or associated directional derivative kernel) and were accelerated through GPyTorch [8] on a single GPU. Code is available at `https://github.com/mishapadidar/GP-Derivatives-Variational-Inference`.

### 5.1 Synthetic functions

In this section, we consider low-dimensional regression with derivatives on test functions including Branin (2D), SixHumpCamel (2D), Styblinksi-Tang (2D) and Hartmann (6D) from [32], a modified 20D Welch test function [1] (Welch-m)[1], and a 5D sinusoid $f(x) = \sin(2\pi||x||^2)$ (Sin-5). We compare variational GPs without derivatives (SVGP, PPGPR) to variational GPs with derivatives (GradSVGP, GradPPGPR), exact GPs with derivatives (GradGP), non-variational GPs with derivatives (DSKI), and variational GPs with $p = 2$ directional derivatives per inducing point (DSVGP2,DPPGPR2). Exact and variational GPs with full derivatives are only tractable in low-dimensional settings due to the scalability issues mentioned in sections 3 and 4.1. Therefore, to apply GradSVGP and GradPPGPR on the 20D Welch-m function, we first perform dimension reduction onto a low dimensional active subspace [2], similar to [5]. To show the limitation of GradSVGP and GradPPGPR, we modified the Welch function to have a low-quality low-dimensional active subspace.

In this low-dimensional setting, we find that variational GPs with directional derivatives, DSVGP2 and DPPGPR2, perform comparably to the methods that incorporate full derivatives (DKSI, GradSVGP, GradPPGPR, GradGP); see Table 1. In Figure 1 we compare the negative log likelihood of each method as the inducing matrix size grows on the Sin-5 and Hartmann test functions. We find that DSVGP2 and DPPGPR2 often outperform other methods due to their ability of incorporating derivative information while only modestly increasing the inducing matrix size.

### 5.2 Implicit Surface Reconstruction

In order to further validate the fidelity of our method's derivative modeling, we consider the surface reconstruction task considered in [5]. We compare to DSKI with the goal of achieving comparable performance, as DSKI is nearly exact for this problem. In Figure 2, we reconstruct the Stanford Bunny by training DSVGP with $p = 3$ inducing directions for 1200 epochs and DSKI on 11606 noisy observations of 34818 locations and corresponding noise-free surface normals (gradients of the bunny level sets). DSVGP smoothly reconstructs the bunny and is comparable to DSKI.

### 5.3 Training Graph Convolutional Neural Networks with Bayesian Optimization

In this section, we demonstrate the full scalability of our approach by training the $D = 4035$ parameters of a two layer graph convolutional neural network (GCN) [16] on the node classification

---

[1]The Welch test function has intrinsically a 6D active space. We modified it to have a low-quality 6D active subspace and to show the limitation of GradSVGP and GradPPGPR.

| | Branin | | Camel | | StyTang | | Sin-5 | | Hartmann | | Welch-m | |
|---|---|---|---|---|---|---|---|---|---|---|---|---|
| | RMSE (1e-3) | NLL | RMSE (1e-3) | NLL | RMSE (1e-3) | NLL | RMSE (1e-1) | NLL | RMSE (1e-1) | NLL | RMSE (1e-2) | NLL |
| SVGP | 1.45 | -3.12 | 5.28 | -2.95 | 3.64 | -3.06 | 6.64 | 0.99 | 1.02 | -0.69 | 16.20 | -0.39 |
| PPGPR | 1.60 | -3.21 | 6.46 | -3.10 | 4.64 | -3.17 | 4.35 | 0.35 | 3.02 | -1.28 | 18.08 | -0.56 |
| GradGP | 15.4 | -0.87 | 25.1 | -0.22 | 44.4 | -0.82 | **2.59** | **-.23** | **0.50** | -0.74 | 16.3 | -0.38 |
| GradSVGP | 0.35 | -3.65 | 2.09 | **-3.62** | 1.00 | -3.65 | 4.85 | 2.31 | 2.08 | 0.59 | 18.94 | 42.82 |
| GradPPGPR | 0.67 | -3.32 | 23.1 | -3.14 | 2.91 | -3.30 | 4.83 | 0.37 | 3.95 | -1.16 | 18.92 | -0.25 |
| DSVGP2 | **0.29** | -3.10 | **1.82** | -2.50 | **0.86** | -2.97 | 3.03 | 1.87 | 0.92 | -0.75 | **3.74** | **-0.74** |
| DPPGPR2 | 0.47 | -3.32 | 8.43 | -3.24 | 1.75 | -3.31 | 4.30 | 0.05 | 2.69 | **-1.64** | 26.08 | -0.71 |
| DSKI | 0.91 | **-4.47** | 3.85 | -3.00 | 1.59 | **-4.74** | N/A | N/A | N/A | N/A | N/A | N/A |

Table 1: Regression results on Branin (2D), SixHumpCamel (2D), Styblinksi-Tang (2D), Sin-5 (5D), Hartmann (6D) and Welch-m (20D), each with 10000 training and 10000 testing points. Following [5], we train GradGP on $10000/(D+1)$ points. The inducing matrix size is 800 for all variational inducing point methods, while DSKI is trained on 800 inducing points per dimension. See the supplementary material for error bars.

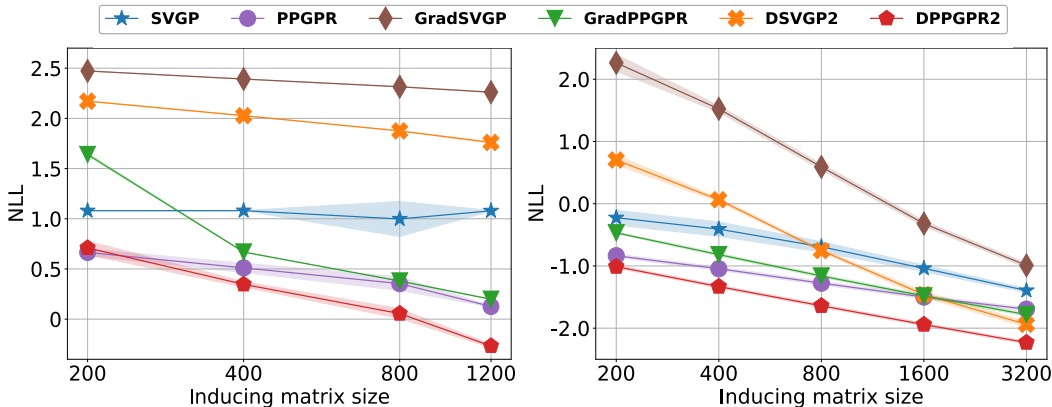

Figure 1: Negative Log Likelihood (NLL) for the various GPs when using different inducing matrix sizes to regress on Sin-5 (Left) and Hartmann (Right). DPPGPR2 often outperforms other methods due to its ability to incorporate derivative information while only modestly increasing the inducing matrix size. DSKI is removed because it does not have comparable matrix size. In both figures, shaded regions correspond to standard errors.

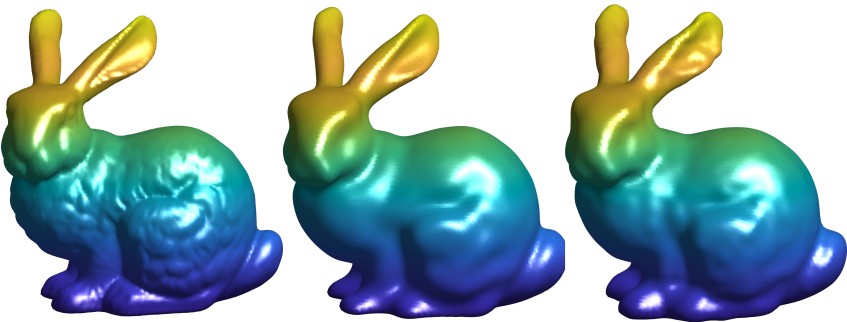

Figure 2: Surface reconstruction of the Stanford bunny: (Left) Original surface, (Middle) DSVGP with 800 inducing points and 3 directions, (Right) D-SKI with $30^3$ inducing grid points.

task of the Pubmed citation dataset [28] using Bayesian optimization. The Bayesian optimization setting compounds the need for scalability, as the GP model must be retrained after each batch of data is acquired. For example, in the last 500 of 2500 optimization iterations with a batch size of 10, a GP must be fit 50 times to datasets with $N(D+1) \approx 2500(4035+1) > 10^6$ combined function and partial derivative labels. Any one of these datasets would be intractable to existing methods for training GPs with gradient observations.

For this experiment, we make no effort to modify the Bayesian optimization routine itself to account for the derivative information (e.g., as in [37]), as this would confound the performance improvements achieved by higher fidelity modelling by incorporating derivative information. Instead, we focus only on swapping out the underlying Gaussian process model. We consider TuRBO [6] as a base Bayesian optimization algorithm which we run with an exact GP, PPGPR, DPPGPR1, DPPGPR2 surrogate models. See supplementary material for $p = 3$ results. We additionally include traditional BO with the Lower Confidence Bound (LCB) [31] acquisition function, Adam [15] and random search. All algorithms were initialized with 400 random evaluations, and the TuRBO variants were run with a batch size of 20 and retrained over 150 steps. Figure 3 summarizes results averaged over 6 trials. We observe that TuRBO with DPPGPR significantly outperform traditional BO and other TuRBO variants. While all Bayesian optimization methods under-perform compared to Adam, we conjecture that this performance gap could be narrowed by incorporating the gradient information into the Bayesian optimization algorithm itself.

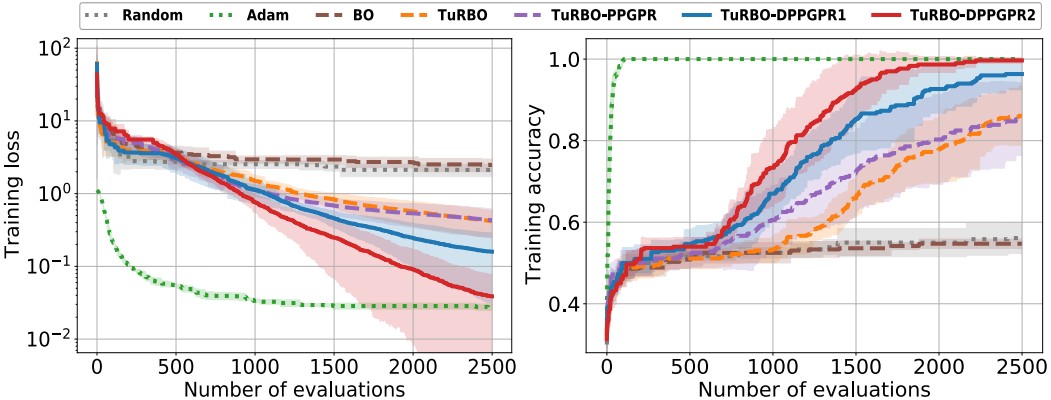

Figure 3: GCN training on the Pubmed dataset: (Left) training loss and (Right) training accuracy. Averaged over 6 trials for all optimizers. Shaded regions correspond to standard errors.

## 5.4 Stellarator Regression

In this experiment, we show the capacity of variational GPs with directional derivatives to extend GP regression with derivatives to massive datasets in a high dimensional settings. We perform regression on $N = 500000$ function and gradient observations gathered from a $D = 45$ dimensional optimization objective function through the FOCUS code [39]: a code for evaluating the quality of magnetic coils for a Stellarator, a magnetic confinement based fusion device for generating renewable energy [12]. The dataset is available upon request.

We compare variational GPs with directional derivatives using $p = 1, 2$ directions (DSVGP1, DPPGPR1, DSVGP2, DPPGPR2) to variational GPs without directional derivatives (SVGP, PPGPR). See supplementary material for $p = 3$ results. While $D$ is too large to use variational or exact GPs with derivatives, we can apply variational GPs with derivatives to a projection of the data set onto a low-dimensional active subspace as in section 5.1. Variational GPs with derivatives trained on reduced datasets of dimension two and three performed poorly compared to all other methods tested. The results of this experiment are shown in Figure 4: variational GPs with directional derivatives significantly enhance regression performance. Even the inclusion of one directional derivative is enough the enhance the predictive capabilities of the regressor. The experiments were averaged over 5 trials, using an Adam optimizer with a multi-step learning rate scheduler and 1000 epochs.

## 5.5 Rover Trajectory Planning

The rover trajectory planning problem [6, 35] is a $D = 200$ dimensional optimization problem with gradients. This experiment validates the use of variational GPs with directional derivatives in Bayesian optimization by leveraging derivative information in a setting where no other methods can. We solve a variant of the rover problem in which the goal is to find an open-loop controller that minimizes the energy of guiding a rover through a series of waypoints in the $xy$-plane. The rover trajectory is integrated over 100 steps at which forces in $x$ and $y$ directions are applied to the rover,

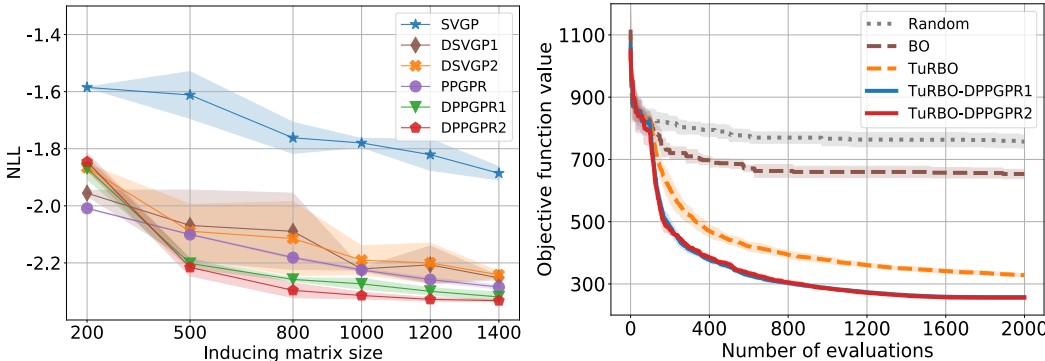

Figure 4: **Stellarator Regression** (Left) and **Rover** (Right). Negative log likelihood of GP variants as the inducing matrix size increases for the $D = 45, N = 500000$ Stellarator Regression experiment. Rover (Right) shows the value of the objective function over the course of optimization. In both figures, shaded regions correspond to standard errors.

making a total of $D = 200$ decision variables. We compare the performance of TuRBO, TuRBO with DPPGPR using $p = 1, 2$, traditional Bayesian optimization with the LCB acquisition function, and random search. All algorithms were initialized with a 100-point experimental design, and the TuRBO variants were run with a batch size of 5, and retrained over 300 steps. The results, averaged over 5 trials, are summarized in Figure 4. We observe that the TuRBO variants that leverage derivative information outperform the other algorithms almost immediately.

## 5.6 Ablation Study Over Number of Inducing Directions

Increasing the number of inducing directions $p$, should increase model flexibility and allow for the model to capture the full set of derivative information. However, as inducing directions are distinct to each inducing point, and are trainable parameters, each inducing directional derivative can encode information from multiple partial derivative directions. Thus variational GPs with directional derivatives may only need a small set of inducing directions to accurately encode the full set of derivative information. In table 2 we show the effect of increasing $p$ on the performance of DPPGPR in the GCN Bayesian optimization (Section 5.3), stellarator regression (5.4), and rover Bayesian optimization (5.5) experiments. We find that using $p = 1$ is usually sufficient for capturing much

| | GCN | | Stellarator | | Rover |
|---|---|---|---|---|---|
| | Loss (1e-1) | Accuracy (%) | RMSE (1e-2) | NLL | Objective |
| PPGPR | 4.31 | 85.28 | 2.43 | -2.22 | - |
| DPPGPR1 | 1.59 | 96.33 | 2.39 | -2.27 | 258.6 |
| DPPGPR2 | 0.39 | 99.67 | 2.31 | -2.31 | 255.8 |
| DPPGPR3 | 9.65 | 98.89 | 2.32 | -2.31 | 253.0 |

Table 2: The effect of increasing the number of inducing directions $p$ on the performance of variational GPs with directional derivatives on the GCN Bayesian optimization, stellarator regression, and rover trajectory planning tasks. PPGPR was run with 1000 inducing points and DPPGPR$p$ was run with $1000/(p + 1)$ inducing points to ensure equivalent computational complexity. Increasing $p$ from zero to one shows a significant improvement, but performance improvement diminishes after that.

of the benefit of derivative information, and that increasing $p$ can improve performance but at diminishing returns. See supplementary material for more $p = 3$ results on the GCN, Stellarator, and Rover tasks.

## 5.7 UCI Regression

Increasing the number of inducing points for Gaussian process models often results in diminishing returns on final model performance, with $500 \leq M \leq 2000$ often proving sufficient [9, 26, 13, 21]. This saturation is likely due in part to the ability of sparse Gaussian processes to represent variation

in the data, but may also be due to increasingly challenging optimization dynamics as more inducing points are added.

One hypothesis worth exploring is that, in some cases, it may be beneficial to augment a smaller set of inducing points with additional descriptive variables rather than to simply increase the number of inducing points. To that end, we test our method on a number of UCI benchmark regression datasets *for which no derivative information is available*. To accomplish this, a variational GP with directional derivatives is initialized with inducing directional derivatives as normal, but during training, minibatches of data only correspond to function observations. In other words, rows of the matrix $\overline{\mathbf{K}}_{XZ}$ never correspond to partial derivative observations because there are none, however columns of $\overline{\mathbf{K}}_{XZ}$ and rows/columns of $\overline{\mathbf{K}}_{ZZ}$ still correspond to inducing directional derivatives as applicable. No changes to the inference procedure are needed.

We test on a number of UCI datasets [4]: Elevators (D=18 N=16599), Kin40k (D=8, N=40000), Energy (D=8, N=768), Protein (D=9, N=45730), Kegg-Directed (D=20, N=53414). We compare variational GPs without derivatives (SVGP, PPGPR) to variational GPs with $p = 1$ directional derivative per inducing point (DSVGP1, DPPGPR1). To ensure that methods have equivalent time complexity, we fix the inducing matrix size to 800. Thus, DSVGP1/DPPGPR1 use 400 inducing points, so that $400 \times (p + 1) = 800$, while SVGP/PPGPR use a full 800 inducing points. We use an 80-20 train-test split for all experiments, and train for 800 epochs. Interestingly, the results in Table 3 show that learning derivative information can improve prediction performance.

| | Elevators | | kin40k | | Energy | | Protein | | Kegg-directed | |
|---|---|---|---|---|---|---|---|---|---|---|
| | RMSE | NLL | RMSE | NLL | RMSE | NLL | RMSE | NLL | RMSE | NLL |
| SVGP | 0.391 | 0.480 | 0.163 | -0.302 | 0.231 | -0.029 | 0.680 | 1.03 | 0.0870 | -1.028 |
| DSVGP1 | **0.379** | 0.449 | **0.134** | -0.545 | **0.087** | **-1.05** | **0.654** | 0.998 | **0.085** | -1.048 |
| PPGPR | 0.407 | 0.404 | 0.274 | -0.874 | 0.252 | -0.899 | 0.709 | 0.928 | 0.0915 | -1.57 |
| DPPGPR1 | 0.391 | **0.390** | 0.264 | **-1.105** | 0.251 | -0.936 | 0.695 | **0.892** | 0.088 | **-1.643** |

Table 3: Variational GPs with no derivative modelling (SVGP, PPGPR) versus with (DSVGP1, DPPGPR1) on UCI benchmark datasets for which no derivative information is available. Bold entries indicate the best method in the column. See the supplementary material for error bars.

# 6 Discussion

Augmenting GPs with derivative information can significantly improve their predicitive capabilities but at a significant $O(N^3D^3)$ cost. We introduce a novel method for achieving fully scalable — scalable in $N$ and $D$ — GPs with derivative information by leveraging stochastic variational approximations. The resulting model reduces the cost of training GPs with derivatives to $O(M^3p^3)$ time per iteration of training, where $M \ll N$ and $p \ll D$. A practical limitation of our method is that $M, p$ must be small enough for fast computations, which is not a reasonable assumption in very high dimensional problems. Through a series of synthetic experiments and a surface reconstruction task, we demonstrate the quality of our approximate model in low dimensional settings. Furthermore, we demonstrate the full scalability of our model through training a graph convolutional neural network using Bayesian optimization, in addition to performing regression on a large scale Stellarator fusion dataset with derivatives. Lastly, we show that our methods can even have benefit in the regression setting when no derivative information is available in the training set, by including a new avenue to encode information. While this last result is a surprising benefit of GPs with derivatives, it is not well understood and is thus a good direction for future study. While our method may make GPs more accessible to practitioners and researchers for calibrating uncertainty estimates, the fundamental assumption that the data is drawn from a GP may be flawed, leading to poor uncertainty estimates and a lack of robustness altogether. Researchers and practitioners should take care to understand the reliability of the GP model in their setting rather than relying faithfully on a black-box approach.

# 7 Acknowledgements

We acknowledge support from Simons Foundation Collaboration on Hidden Symmetries and Fusion Energy and the National Science Foundation NSF CCF-1934985, and NSF DMS-1645643.

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
