# Supplementary Material: Scaling Gaussian Processes with Derivative Information Using Variational Inference

**Misha Padidar[1], Xinran Zhu[1] Leo Huang[1], Jacob R. Gardner[2], David Bindel[1]**
[1]Cornell University, (map454, xz584, ah839, bindel)@cornell.edu
[2]University of Pennsylvania, jacobrg@seas.upenn.edu

## 1  Results for synthetic functions

|  | Branin | | Camel | | StyTang | | Sin-5 | | Hartmann-6 | | Welch-20 | |
|---|---|---|---|---|---|---|---|---|---|---|---|---|
|  | RMSE | NLL | RMSE | NLL | RMSE | NLL | RMSE | NLL | RMSE | NLL | RMSE | NLL |
|  | (1e-4) | (1e-3) | (1e-3) | (1e-2) | (1e-3) | (1e-2) | (1e-2) | (1e-2) | (1e-2) | (1e-2) | (1e-3) | (1e-2) |
| SVGP | 0.67 | 2.54 | 0.27 | 1.89 | 0.70 | 0.41 | 10.1 | 17.4 | 0.60 | 9.26 | 1.61 | 0.70 |
| PPGPR | 0.50 | 0.52 | 0.46 | 0.55 | 0.21 | 0.47 | 1.29 | 6.80 | 1.98 | 3.11 | 6.18 | 2.39 |
| GradGP | 9.12 | 7.82 | 1.75 | 0.48 | 2.48 | 0.79 | 0.41 | 0.41 | 0.27 | 0.88 | 0.73 | 0.30 |
| GradSVGP | 1.64 | 2.70 | 0.19 | 0.75 | 0.23 | 0.30 | 0.93 | 0.14 | 0.11 | 5.46 | 2.29 | 17.6 |
| GradPPGPR | 2.64 | 1.93 | 16.20 | 10.8 | 1.35 | 1.43 | 0.02 | 0.06 | 1.67 | 2.40 | 2.15 | 1.11 |
| DSVGP2 | 0.87 | 8.29 | 0.21 | 2.61 | 0.11 | 2.00 | 0.60 | 0.25 | 0.25 | 2.69 | 1.44 | 0.81 |
| DPPGPR2 | 1.70 | 0.96 | 2.35 | 2.13 | 0.59 | 0.49 | 1.38 | 4.86 | 1.46 | 2.72 | 4.32 | 1.64 |
| DSKI | 0.25 | 10.8 | 0.20 | 33.6 | 0.03 | 22.0 | N/A | N/A | N/A | N/A | N/A | N/A |

Table 1: Standard errors for regression results on synthetic experiments, i.e. Table 1 in the main paper.

The following plots show detailed results for the synthetic experiments. In each figure we show the Root Mean Squared Error (RMSE) and Negative Log Likelihood (NLL) versus the inducing matrix size. We compare Variational GPs (SVGP, PPGPR) to Variational GPs with Derivatives (GradSVGP, GradPPGPR) and Variational GPs with $p = 2$ Directional Derivatives (DSVGP2, DPPGPR2). The number of inducing points is varied across methods in order to keep the computational complexity equivalent. If SVGP and PPGPR used $M_0$ inducing points then DSVGP2, DPPGPR2 used $M = M_0/(p+1) = M_0/3$ inducing points and GradSVGP, GradPPGPR used $M = M_0/(D+1)$ inducing points. All methods were trained using the Adam optimizer. Methods were implemented using GPyTorch, and run with a single GPU

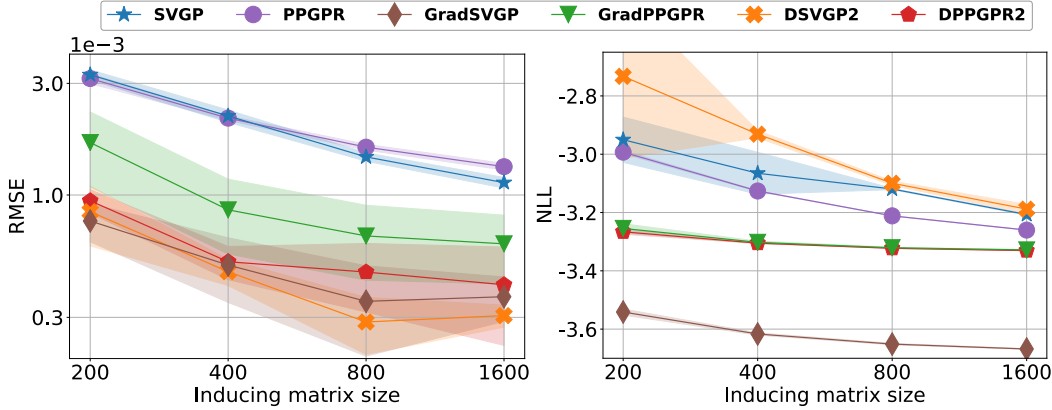

Figure 1: The root mean squared error (RMSE) and the negative log likelihood (NLL) for the various GPs when using different inducing matrix sizes to regress on Branin. Averaged over 5 runs. DSKI is removed because it does not have comparable matrix size.

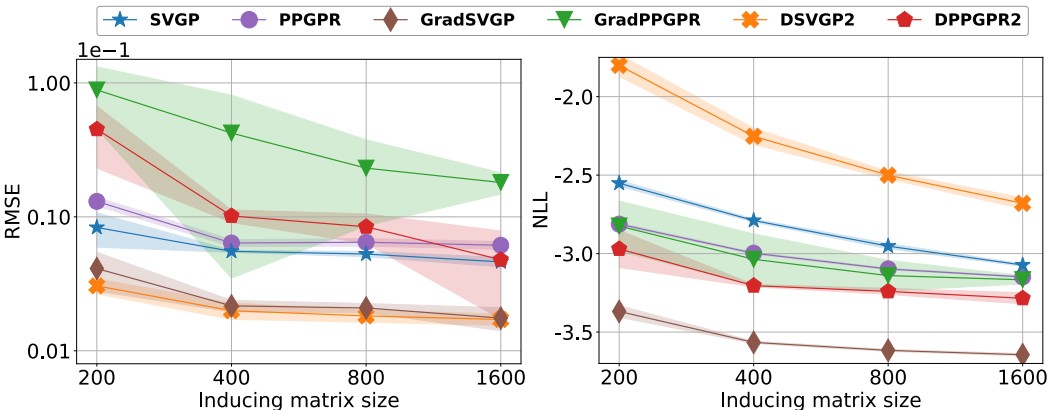

Figure 2: The root mean squared error (RMSE) and the negative log likelihood (NLL) for the various GPs when using different inducing matrix sizes to regress on SixHumpCamel. Averaged over 5 runs. DSKI is removed because it does not have comparable matrix size.

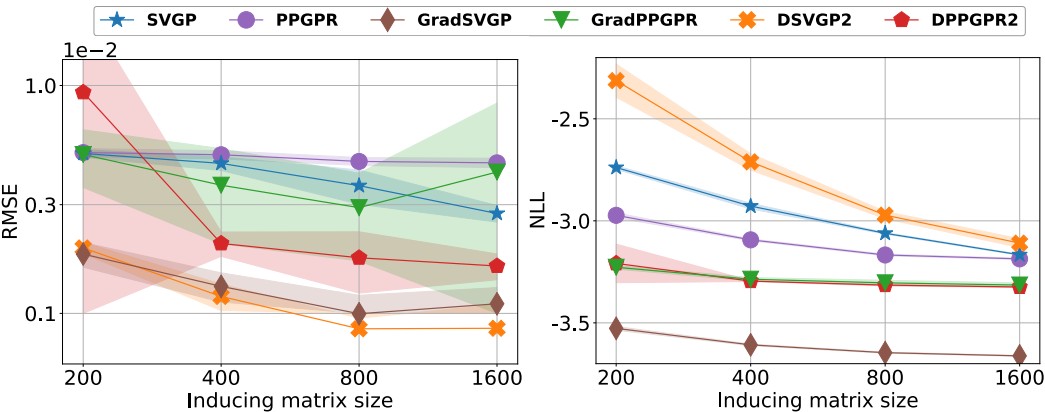

Figure 3: The root mean squared error (RMSE) and the negative log likelihood (NLL) for the various GPs when using different inducing matrix sizes to regress on Styblinksi-Tang. Averaged over 5 runs. DSKI is removed because it does not have comparable matrix size.

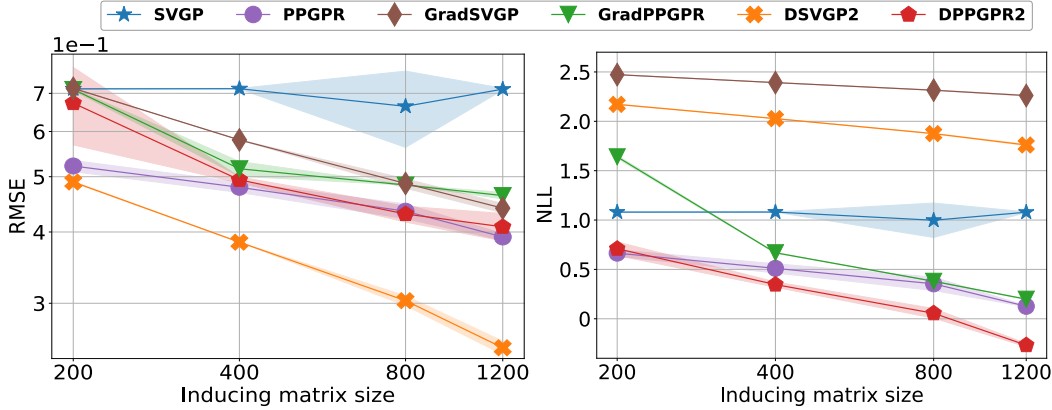

Figure 4: The root mean squared error (RMSE) and the negative log likelihood (NLL) for the various GPs when using different inducing matrix sizes to regress on Sin-5. Averaged over 5 runs. DSKI is removed because it does not have comparable matrix size.

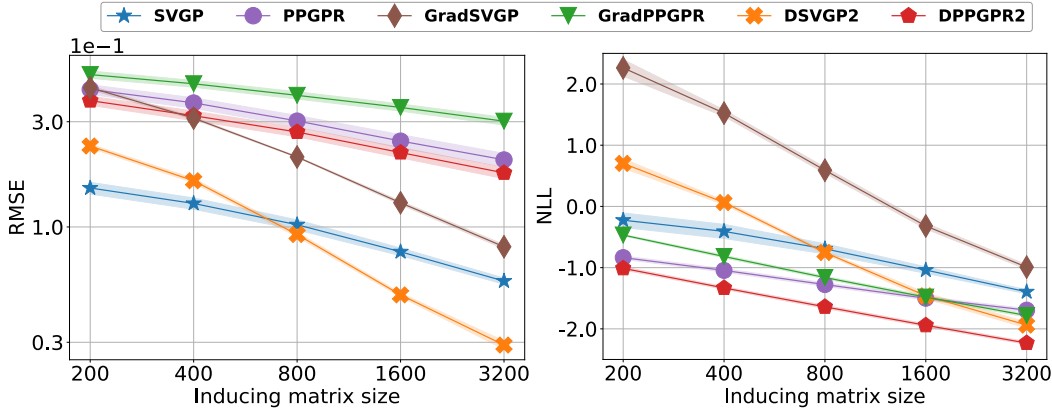

Figure 5: The root mean squared error (RMSE) and the negative log likelihood (NLL) for the various GPs when using different inducing matrix sizes to regress on Hartmann. Averaged over 5 runs. DSKI is removed because it does not have comparable matrix size.

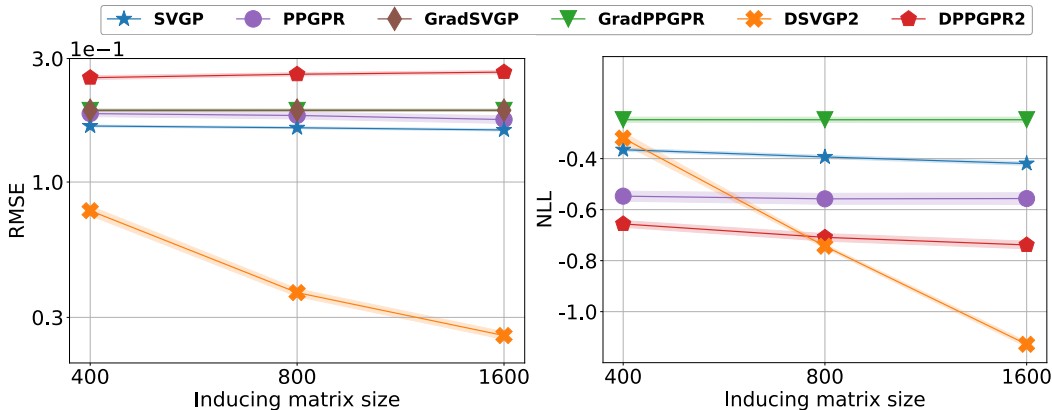

Figure 6: The root mean squared error (RMSE) and the negative log likelihood (NLL) for the various GPs when using different inducing matrix sizes to regress on Welch-m. Averaged over 5 runs. DSKI is removed because it does not have comparable matrix size.

## 2 Results with shared inducing directions

In this section we test the performance of variational GPs with directional derivatives when the set of inducing points learns a set of $p$ *shared* inducing directions, rather than each inducing point having its own distinct set of $p$ directions. We abbreviate these methods as DSVGP-Shared$k$ and DPPGPR-Shared$k$, with $k$ indicating the number of inducing directions. We compare the performance to variational GPs without derivatives (SVGP, PPGPR) and variational GPs with distinct directional derivatives (DPPGPR, DSVGP) on the stellarator regression task and the graph convolutional neural network (GCN) bayesian optimiztion task, see main paper for a description of these experiments. On the GCN bayesian optimization task TuRBO-DPPGPR-Shared$k$, and TuRBO-DSVGP-Shared$k$ indicate the variants of TuRBO that leverage use DPPGPR-Shared and DSVGP-Shared as the GP surrogate.

Note that if $M_0$ inducing points are used with $p$ shared inducing directions, the inducing matrix is still $M \times M$ where $M = M_0(p+1)$. This is equivalent to the inducing matrix size for $M_0$ inducing points and $p$ *distinct* inducing directions. As the complexity of variational GPs with directional derivatives is $O(M^3)$, sharing directions does not improve complexity. Furthermore it significantly reduces model flexibility as the number of inducing parameters is significantly lower than if directions are distinct.

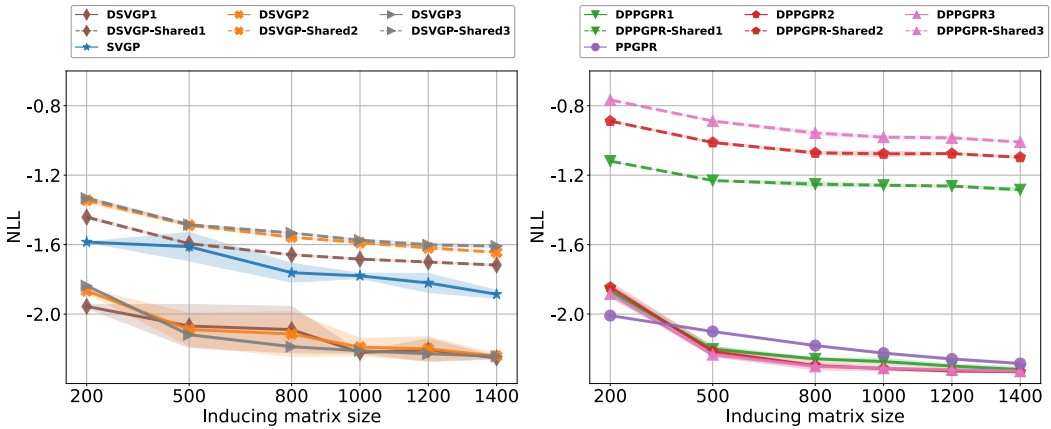

Figure 7: Negative log likelihood of GP variants as the inducing matrix size increases for the $D = 45, N = 500000$ Stellarator Regression experiment. Shaded regions correspond to standard errors. We find that for the same computational complexity (inducing matrix size) sharing inducing directions is inferior to distinct inducing directions, likely due to the lack of model flexibility in this high dimensional problem.

## 3 Standard Error for UCI Regression Experiments

Table 2 shows the standard errors for the UCI Regression experiments in the main paper. See the main paper for a description of the experiments.

| | Elevators | | kin40k | | Energy | | Protein | | Kegg-directed | |
|---|---|---|---|---|---|---|---|---|---|---|
| | RMSE | NLL | RMSE | NLL | RMSE | NLL | RMSE | NLL | RMSE | NLL |
| SVGP | 0.0011 | 0.0028 | 0.0006 | 0.0011 | 0.0057 | 0.0237 | 0.0028 | 0.0037 | 0.0007 | 0.0099 |
| DSVGP1 | 0.0027 | 0.0070 | 0.0008 | 0.0035 | 0.0023 | 0.0214 | 0.0030 | 0.0041 | 0.0008 | 0.0088 |
| PPGPR | 0.0051 | 0.0080 | 0.0027 | 0.0027 | 0.0069 | 0.0274 | 0.0025 | 0.0050 | 0.0014 | 0.0142 |
| DPPGPR1 | 0.0028 | 0.0077 | 0.0059 | 0.0066 | 0.0043 | 0.0269 | 0.0025 | 0.0062 | 0.0015 | 0.0084 |

Table 2: Standard Errors for Variational GPs with no derivatives (SVGP, PPGPR) and Variational GPs with $p = 1$ directions (DSVGP1, DPPGPR1) on UCI benchmark regression datasets for which no derivative information is available. See the main paper Table 3 for regression performance results.

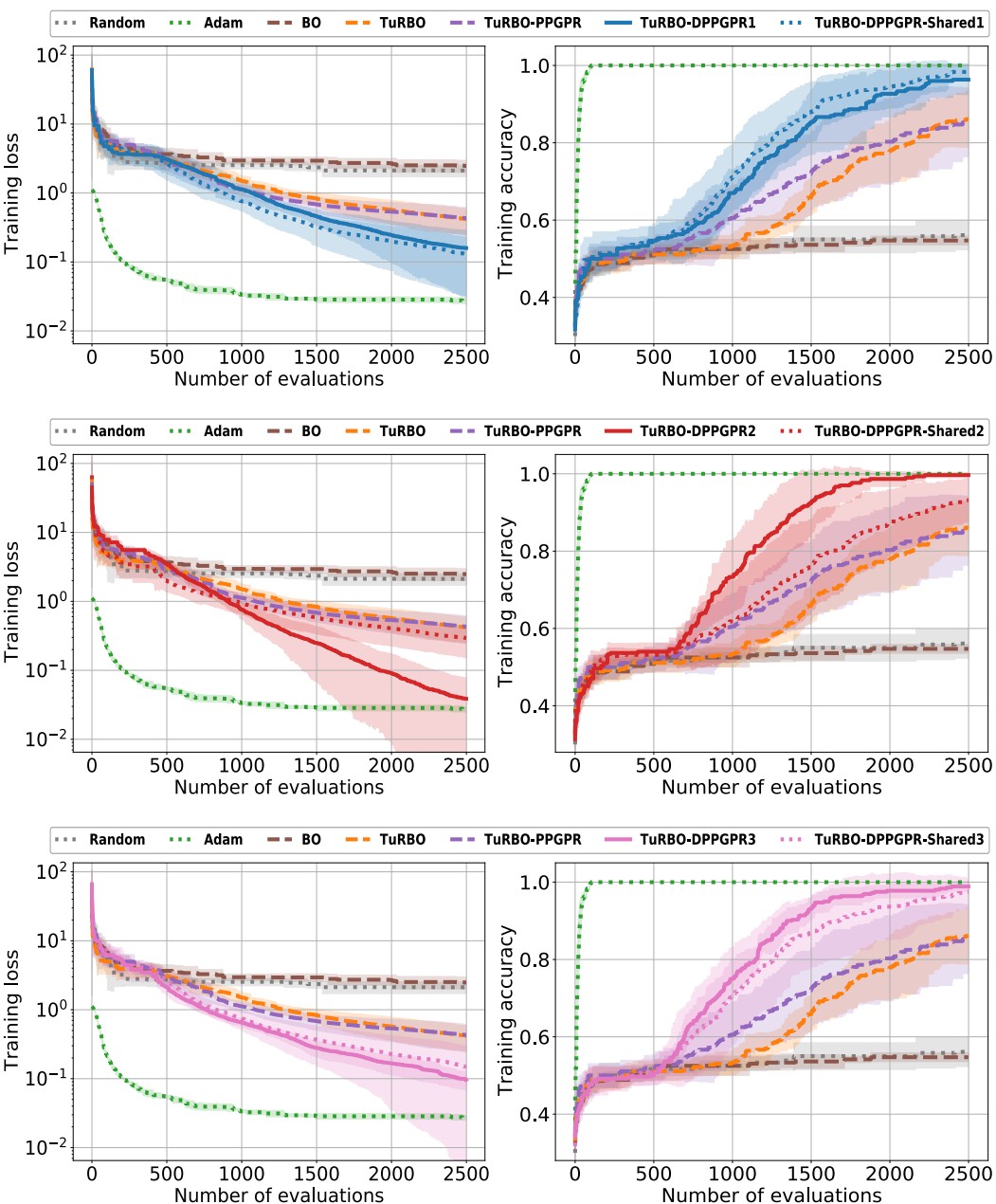

Figure 8: GCN training on the Pubmed dataset: (Left) training loss and (Right) training accuracy. Averaged over 6 trials for all optimizers. Shaded regions correspond to standard errors. From top to bottom, a different number of inducing directions are used: $p = 1, 2, 3$ respectively. We find that using shared inducing directions still provides a benefit over incorporating no derivative information, but does not perform as well as using distinct inducing directions.