# OpenReview forum: "Scaling Gaussian Processes with Derivative Information Using Variational Inference"
_NeurIPS.cc/2021/Conference — NeurIPS 2021 Poster_

### Official Review · Reviewer_TvZh · 2021-07-10

**Rating:** 7
**Confidence:** 4

**Summary:**

In a spirit similar to the SVGPs which sparsify the training data, the authors propose to augment the SVGP framework with derivative information by sparsifying the derivative data with *inducing directional derivatives*. By virtue of these modeling choices, the resulting algorithm scales favorably independent of data size as in SVGPs, and now additionaly also w.r.t. input dimension. Empirically, the authors show much better scalability in both high-N and high-D than existing work.

**Limitations And Societal Impact:**

Yes.

**Main Review:**

Overall, the paper checks all the boxes in terms of writing and narrative framing, presenting a clean linear story. The contribution is important, given that it brings a general approach to augment variational GP models with derivative information via PPGPR to allow heteroscedasticity. The choice of using directional derivatives is natural in hindsight, making this contribution even more important.

One setting which seems to be missing is the choice of using a fixed set of inducing directions for all inducing inducing points, instead of picking unique directions for every inducing point. Intuitively, is this choice the natural first step? Would the flexibility of the model suffer if we started with directions shared across all inducing points?

Section 5.6 on using DPPGPR even when no derivative information is available is interesting, but not entirely unclear. How does the model in Eq. $(6)$ change if we do not have derivative information. This does not appear to have been spelled out clearly, or isn't obvious.

**Time Spent Reviewing:**

5

---

> ### Author Response · Authors · 2021-08-05
> **Author response**
>
> Thank you for your careful review of the paper and suggestions.
>
> **Shared inducing directions**
>
> We agree that a shared set of inducing directions amongst all inducing points would be a natural intermediate model. Indeed, this was our initial choice when developing the method. However, in developing the method we realized that under either choice (shared vs independent directions), the size of the kernel matrices remains the same, giving both methods a computational complexity of O(M^3p^3), making the added flexibility of having independent directions per inducing point essentially free from a computational perspective. This cheap flexibility incentivized us to use inducing points with distinct directions. We will add experiments to the supplementary results that demonstrate the empirical performance loss that we noticed from fixing a shared set of inducing directions. For example, on stellerator, DSVGP1-shared is 0.41 nats NLL worse than DSVGP1, and DPPGPR1-shared is 0.91 nats of NLL worse than DPPGPR1, with similar results for p=2.
>
> **Adapting the model to no derivative observations**
>
> We agree that section 5.6 will be improved by more carefully describing how to adapt the model to the setting where derivative observations are unavailable, and we will more thoroughly describe this extension. The adaptation is mechanically fairly straightforward. In the setting with access to derivative information, minibatches of data can contain labels (e.g., (x, y) pairs) and/or arbitrary subsets of partial derivatives (e.g., (x, \partial_{j} y) pairs), resulting in matrices K_{XZ} where some rows correspond to the covariance between inputs x and inducing points/directional derivatives and some rows correspond to the covariance between true partial derivatives and inducing points/directional derivatives. In the setting with no derivative observations, minibatches simply never contain partial derivative inputs (e.g, (x, \partial_{j} y) pairs). Thus, rows of K_{XZ} always correspond to inputs, while columns of K_{XZ} and entries of K_{ZZ} can still correspond to inducing directional derivatives. Our code release to reproduce this experiment will likely lend additional clarity beyond the text, as well.

---

### Official Review · Reviewer_g1G4 · 2021-07-14

**Rating:** 5
**Confidence:** 4

**Summary:**

This paper presents a new method to enable Gaussian processes with derivatives to deal with large and high-dimensional datasets. Several synthetic and real-world experiments are conducted in evaluating the proposed method.

**Ethical Concerns:**

N.A.

**Limitations And Societal Impact:**

refer to the detailed comments above.

**Main Review:**

Originality and Significance: To be honest, I am a little bit disappointed when reading the technical part and experiments. Since two key ingredients discussed in the introduction:
1. heteroscedastic noise
2. principal to select $p$

I am expecting the authors can address 1. what is the advantages of heteroscedastic noise rather than i.i.d noise? 2. how will the model perform with different $p$.

It is natural that selecting a small number of $p$ can reduce the time complexity, however, the authors did not really mention how to determine $p$ for different tasks? PCA, t-sne, or even gplvm can do this dimensionality reduction right? I am expecting more discussion relating to derivative and dimensionality reduction. This should be the golden apple of this paper.

Moreover, Sec 4.1 seems to be simple adaption of VSGP to VSGP with derivatives, is there any difficulty in adopting VI? Any other benifits in adopting VI other than reducing time complexity?

The two aspects above are not extensively discussed and evaluated in the paper which makes the contribution incremental.

Quality & Clarity: The writing of this paper is really smooth and easy to follow. I really appreciate the efforts the authors put in polishing this paper.

**Time Spent Reviewing:**

6

---

> ### Author Response · Authors · 2021-08-05
> **Author response**
>
> Thank you for pointing out some areas that we agree would benefit from additional clarification. We address these points below, and will be sure to incorporate this discussion into the final paper.
>
> **Regarding the selection of p**
>
> As with M for SVGP, in principle p should simply be set as high as possible within a given computation budget, as this results in less approximate derivative modelling. In practice, it’s better to consider the inducing matrix size R=M(p+1) as the computational bottleneck rather than M or p independently, as we do in our experiments because each inducing point has its own inducing directions (leading to M * p total inducing directions). Increasing p while reducing M to maintain fixed R may result in modelling fewer directions overall in a given region by having fewer inducing points nearby, which explains the diminishing returns of increasing p too much. While we found p=1 or p=2 to be sufficient for the tasks we considered, we are happy to add p=3 to all experiments. On the stellarator regression task, we get the following NLLs with R fixed to 1400: DPPGPR1=-2.21, DPPGPR2=-2.24, DPPGPR3=**-2.32**. It's unlikely that one would use much larger values of p in practice due to the scaling cost of inverting K_{ZZ}. In addition to likely being unnecessary, it is also not straightforward to select p with dimensionality reduction techniques like PCA, as these would give global derivative subspaces, whereas we use different directional derivatives for each inducing point.
>
> **Regarding the selection of V using pca, etc**
>
> We learn the inducing directions V_i by directly maximizing the ELBO with respect to them, much in the same way that SVGP learns the inducing point locations z_i by maximizing the ELBO. We view this as the most principled mechanism for learning V, as it is standard practice for hyperparameters and inducing parameters in variational GP models. That said, initialization of the inducing locations e.g via k-means is known to improve performance, and it is likely that there is an analogous procedure to aid in selection or initialization of the inducing directions. Applying PCA and other mechanisms to select directions is surprisingly not straightforward however, as a naive global linear subspace would require using the same set of inducing directions at every inducing point (see response to Reviewer TvZh for empirical performance loss of sharing directions). While learning V through the ELBO as we do is the most principled analog of how other parameters are handled, we believe it would be interesting to explore how existing dimensionality reduction techniques like those you suggest could be adapted to create initialization schemes.
>
> **Regarding heteroskedasticity**
>
> We assume you are referring to our discussion in lines 90-95. While modelling heteroskedasticity is not a critical component of our core contribution for scalability (note that we include results on both DPPGPR and DSVGP), changing from DSVGP to DPPGPR is a one line of code change that accomplishes two goals with derivative modelling. First, a single global noise scalar \sigma^{2}_{n} may not be appropriate in the derivative setting for the simple reason that the function and its derivatives may have different observational noise -- modelling derivatives calls for additional noise modelling in the same way that multi-task Gaussian processes do. Second, derivative observations may be more likely to have input-dependent noise than the function: for example, a function with constant observational noise may have lower derivative noise near a critical point where the derivative is very small independent of the magnitude of the function f.
>
> **Regarding section 4.1**
>
> As Reviewer TvZh points out, we believe that section 4.2 is a natural scalable evolution of the model proposed in section 4.1. This section therefore serves a critical narrative purpose of both better motivating section 4.2 and making it easier to explain as a modification to a “first-attempt” SVGP with derivatives method. While we agree that the results of our derivation in 4.1 lead to an inevitable outcome, we are unaware of prior work that considers derivative modelling in the SVGP setting. It is therefore still important to formally write down, especially to highlight that the ELBO decomposes as a sum over both labels and individual partial derivatives.
>
> **"Which makes the contribution incremental."**
> We would like to emphasize that the tasks considered in sections 5.3-5.5 are well beyond the computational capabilities of existing work for conditioning Gaussian processes on derivatives. For some of these applications, DSKIP and exact GPs with derivatives would require more than 100 terabytes of GPU memory.

---

### Official Review · Reviewer_VwPg · 2021-07-16

**Rating:** 5
**Confidence:** 4

**Summary:**

The authors propose a scalable inference method for GP with derivatives. Previous methods could only work for large data in low-dimensions or small data in high-dimensions. They solve the issues by using the inducing points and directional derivatives. They compare with other alternatives on multiple datasets and show that scalable GP with directional derivatives has the best performance.



**Limitations And Societal Impact:**

1. mu_f(x_i) is not defined before or after the use in eq. 4.

2. How to construct kernel matrices in eq. 12 from eq. 11?

3. How to select the directions for v? Especially with large D. I assume mapping high-dimensional derivatives down to a scalar with v and p=1,2 would be quite tricky.

4. I don't understand this sentence very much "as this would confound the performance improvements achieved by higher fidelity modeling by incorporating derivative information." Could you explain it? I'm confused that if the derivative information is not considered in BO, what is \nabla f in this example?

5. Why DSKI have N/A in table 1?

6. In sec 5.2, how many inducing points, i.e., M, are used with DSVGP? DSKI on 11606 observations of 34818 locations. What are 11606 and 348181?

7. Since DSKI claims that it could scale to high dimensions, it would still be helpful to include DSKI for applications with large D, like sec 5.3 and 5.4.

8. Flip the order of the two figures in figure 4 since stellarator is introduced before rover.

9. It's a bit confusing to arrange the experiments with the current order: 5.1 regression, 5.2 surface reconstruction, 5.3 BO, 5.4 regression, 5.5 BO, 5.6 regression without derivative information. It feels like the comparable methods switch between different groups. This makes the logic hard to follow. It might be helpful to have a table summarizing what application the experiment is and whether it's large N or large D or both.








**Main Review:**

The directional derivative idea to map high-dimensional derivative kernels down to a scalar is novel and seems to work practically well. However, I'm still a bit skeptical about the choice of directions which could have a potential effect I assume. But this is not analyzed in the paper at all. The paper needs to provide better intuitions, especially in the experiment part to convince readers about that. The paper needs more edits to make it clear and easy to follow.


**Time Spent Reviewing:**

2 hours

---

> ### Author Response · Authors · 2021-08-05
> **Author response**
>
> Thank you for pointing out some opportunities for us to provide additional clarification for key details. We will thoroughly incorporate the discussion below into the text to improve clarity about the method.
>
> **Regarding the choice of inducing directions V**
>
> As mentioned in section 4.2, we learn the inducing directions V_i by maximizing the ELBO with respect to them jointly with all other hyperparameters, much in the same way that SVGP learns the inducing point locations z_i by maximizing the ELBO. We view this as the most principled choice, as it is standard practice for hyperparameters and inducing parameters in variational GP models. *Because this aspect of our method is so critical, we will solve the clarity issue that lead to this confusion by introducing a paragraph heading “Learning inducing directions”' so that this information is separated out with a bold heading.*
>
> **“I assume mapping high-dimensional derivatives down to a scalar with v and p=1,2 would be quite tricky.”**
>
> We do not reduce derivative information down to a scalar or single direction, even with p=1. Each inducing point z_i has its own set of p learnable inducing directions v_i, leading to M*p different derivatives in M*p different directions considered, for a total of M*p*(d+1) parameters. Inducing points that are close together but have different inducing directions therefore allow the model to capture a potentially much higher than p dimensional subspace: see response to Reviewer TvZh for empirical performance loss of sharing directions. See lines 179-187 for more discussion.
>
> **1,8** Thank you for pointing out these omissions and opportunities for improved clarity. We will correct them.
>
> **2.** The kernel matrices are constructed in O(M^2p^2) time directly from the kernel function given in equation (11) by computing the kernel function for relevant pairs of data points, inducing points, derivatives and inducing direction derivatives. This can be implemented as a standard Kernel in most GP packages -- we will release code.
>
> **4.** Here, we mean that we don’t modify the Bayesian optimization search strategy to account for derivative information beyond the surrogate mode (for example, the acquisition function). Thus, in our BayesOpt experiments, optimization improvements stem solely from improved GP modelling performance, which is the primary focus of our paper. We may even remove this sentence -- our main goal was simply to acknowledge that moving beyond better surrogate modelling (which we do achieve here) and additionally devising better search strategies with access to derivatives is important future work.
>
> **6.**. See the caption for Figure 2. DSVGP used 800 inducing points with p=3 directions, DSKI used 30^3 inducing points.
>
> **5,7.** DSKI does not scale well beyond 2-5 dimensions due to an exponential dependence on D, hence the N/A values. DSKIP utilizes SKIP, which alleviates this to some degree but in practice (as the authors of SKIP point out in section 7 of the SKIP paper) is typically limited to 10-20 dimensions due to the space complexity of computing and storing O(d log d) Lanczos decompositions. Past the experiments in sections 5.1 and 5.2, the settings we consider are vastly beyond the capabilities of either baseline. For example, the stellarator dataset would require at least 103 GB of GPU memory to apply DSKIP, while the GCN training would require 4.3 terabytes.
>
> **9.** We consider 5.1 and 5.2 to primarily be settings that enable us to compare prior work. Sections 5.3 through 5.5 are well beyond the scalability means of all prior work we are aware of. We will clarify the structure of section 5.

---

### Official Review · Reviewer_GftX · 2021-07-16

**Rating:** 7
**Confidence:** 4

**Summary:**

   The manuscripts proposes a method to accelerate GP inference in situations where derivative observations are available and the number of data points N and the data dimension D tends to be large. Extending previous work on stochastic variational approximations, the central idea of the paper is to use a number p of directional derivatives as variational parameters rather than full derivatives which greatly reduces the computational footprint.

   Strengths
   + Computational methods to tackle high D, high N GP inference problems with derivative information available are not fully scalable and hence a relevant research subject.
   + The proposed approach is clear and builds on established techniques.
   + The computational benefits demonstrated in an extensive set of experiments are substantial and convincing.


   Weaknesses
   - Experimental results are without error bars.

**Ethical Concerns:**

No concerns.

**Limitations And Societal Impact:**

Yes.

**Main Review:**

   1) Clarity

   The paper is well written. The notation is concise and the technical content is easily accessible. The experiments are well described.

   2) Originality

   The proposal to use directional derivatives as variational parameters to render GP inference with derivatives more scalable seems novel.

   3) Significance

   Although the setting of GP regression with high N, high D and derivative information available sound like a niche application, the presented experiments show that these settings do exist and could not be tackled before. Beyond this, the scope of the method seems rather limited.

   4) Reproducibility

   The data and the code to run the experiments will be released upon acceptance. Although not available during the review process, it should be possible to reproduce the results.

   5) Empirical analysis

   The manuscript contains a wide range of experiments illustrating various aspects of the method on a number of data sets along with comparative results relative to the relevant baseline methods.

   6) Minor Issues and Typos

   line 197: DSKIP rather than DKIP
   line 206: learning tasks, we perform
   Table 1, caption: Hartmann rather than Hatrmann
   Figure 1, caption: ability to incorporates
   line 256: In this experiment, we show
   line 283: in Figure 4
   line 312: reasonable assumption in very high

**Time Spent Reviewing:**

3

---

> ### Author Response · Authors · 2021-08-05
> **Author response**
>
> Thank you for your careful review and suggestions, we will incorporate your feedback as follows.
>
> We will add error bars to Table 1 and Table 2 in the final draft, and add clarification to the text that the shaded regions currently in figures 1, 3 and 4 correspond to standard errors. Thank you for reading carefully and finding our typos and minor issues. We will make sure to correct those.

---

### Decision · Program_Chairs · 2021-09-27

**Decision:**

Accept (Poster)

**Comment:**

This paper addresses the problem of scalable inference in Gaussian process (GP) regression with derivative information for the general case where both the number of observations (N) and the dimensionality (D) are large. While previous work has addressed the large-N-low-D and low-N-large-D regimes independently, the paper proposes the use of inducing directional derivatives (which play a similar role to that of the inducing variables in standard variationally sparse GP models) rather than full derivatives to derive a scalable variational inference algorithm. A comprehensive set of experiments (on synthetic data, implicit surface reconstruction, training graph convolutional networks with Bayesian optimization, large-scale regression with derivative information, Rover trajectory planning and standard GP regression on UCI datasets) show the benefits of the proposed approach with respect to previous methods.

I believe the contribution of this paper to be significant to the NeurIPS/GP community as it develops a new method for settings where previous work cannot be applied. The reviewers raised several concerns regarding the choice of inducing directions and the use of different directions per inducing point. I believe the authors have addressed these successfully. A limitation of this work is the very small number of directional derivatives (p) used in the experiments (p=1, 2). As the complexity of the proposed algorithm scales as (M^3 p^3), exploring larger p values will necessarily lead to fewer inducing variables (M).